# Prescribing Patterns and Adverse Effects of Semaglutide: A Real-World Comparative Evaluation

**DOI:** 10.3390/healthcare14010035

**Published:** 2025-12-23

**Authors:** Abigail Whorton, Samira Osman, Jaspal Johal, Sarah Baig, Alan M. Jones, Zahraa Jalal

**Affiliations:** 1School of Pharmacy, University of Birmingham, Edgbaston, Birmingham B15 2TT, UK; abbie.whorton1@gmail.com (A.W.); s.s.baig@bham.ac.uk (S.B.); a.m.jones.2@bham.ac.uk (A.M.J.); z.jalal@bham.ac.uk (Z.J.); 2The Dudley Group NHS Foundation Trust, Pensnett Road, Dudley DY1 2HQ, UK; jaspal.johal@nhs.net

**Keywords:** semaglutide, Ozempic, GLP-1 receptor agonists, prescribing patterns, adverse drug reactions, type 2 diabetes, health inequality, pharmacovigilance

## Abstract

**Background**: Semaglutide is a Glucagon-like peptide-1 receptor agonist (GLP-1RA) used in the treatment of type 2 diabetes mellitus (T2DM) and weight management. While its clinical benefits are well established, concerns have emerged over off-label use, underreporting of adverse drug reactions (ADRs), and prescribing disparities. **Aims**: To examine real-world prescribing pattern treatment efficacy and ADRs associated with semaglutide in a socioeconomically deprived United Kingdom (UK) locality, and to compare these with national data. **Methods**: A retrospective service evaluation was conducted using anonymised data from 1403 patients across 42 GP practices under a data share agreement across a place-based group of practices in the West Midlands. National prescribing data were obtained from OpenPrescribing, and ADR data from the Medicines and Healthcare products Regulatory Agency (MHRA) Yellow Card Scheme (01/2020–12/2024). Analyses included demographic trends, treatment efficacy (HbA1c and weight), and socioeconomic comparisons using the Socioeconomic Index for Small Areas (SEISA) deciles. **Findings**: Semaglutide prescribing in the GP surgeries studied peaked in 2022 and declined thereafter, mirroring national trends. Prescribing of semaglutide mirrored the ethnic make-up of the region studied with a notable exception of White British. Mean HbA1c fell by 10.8 mmol/mol and weight by 4.8%. ADR incidence in the population studied (1.85%) exceeded national reporting rates (0.20%). Prescribing was highest in practices serving the most deprived communities. **Conclusions**: Semaglutide is effective in reducing HbA1c and weight in real-world settings. However, ADRs remain underreported. Socioeconomic deprivation was strongly associated with higher prescribing rates. Greater attention to equitable access and pharmacovigilance is warranted.

## 1. Introduction

Glucagon-like peptide-1 receptor agonists (GLP-1RAs) are established treatments for type 2 diabetes mellitus (T2DM), improving glycaemic control through multiple mechanisms, including enhanced insulin secretion, delayed gastric emptying, and appetite suppression [1,2]. Since the approval of exenatide in 2005, several GLP-1RAs—including liraglutide, dulaglutide, and semaglutide—have been licenced in the UK, with semaglutide gaining approval in 2017 [3,4].

Semaglutide is now widely used for both T2DM and weight management, marketed under Ozempic and Wegovy, respectively [5,6]. Its clinical benefits have been demonstrated in the SUSTAIN and STEP trials, with significant reductions in HbA1c and body weight [7,8]. These effects have also led to increasing off-label use, including prescribing for weight loss in non-diabetic populations [9].

However, growing public demand, unregulated non-General Pharmaceutical Council (GPhC) approved online supply, and off-label prescribing have raised safety concerns. Reports include inappropriate use via social media platforms [10], adverse outcomes due to lack of clinical oversight [11], and supply shortages affecting diabetic patients [9]. Additionally, falsified semaglutide products have been identified in circulation [12].

Adverse effects, particularly gastrointestinal, are common with semaglutide, and rare but serious events such as pancreatitis and thyroid cancer have been reported [13,14,15]. Yet pharmacovigilance data remain limited, with most studies relying on US-based systems such as the FDA Adverse Event Reporting System [16,17,18]. UK-specific evidence on real-world adverse drug reactions (ADRs) and prescribing patterns is limited.

Semaglutide is the most frequently used off-label GLP-1 RA; to lower the risk of ADRs, previous studies exploring VigiAccess and EudraVigilance data have suggested that off-label semaglutide use should be carefully monitored and reduced [19,20].

Furthermore, little is known about how prescribing varies by local demographics, ethnicities, and socioeconomic statuses; however, existing studies have suggested that there are potential inequities in prescribing that require further exploration [21]. This is particularly important in areas such as the area included in this study in the West Midlands/England, where over half of the population lives in the most deprived quintiles nationally [22]. Given that deprivation and ethnicity are associated with both diabetes prevalence and treatment outcomes, local data are needed to guide safe and equitable prescribing.

The aim of this research is to examine real-world prescribing pattern treatment efficacy and ADRs associated with semaglutide in a socioeconomically deprived United Kingdom (UK) locality, and to compare these with national data.

### Specific Objectives

1.Identify potential correlations in patient characteristics and incidence of ADRs experienced associated with the use of semaglutide.2.Compare local prescribing patterns of semaglutide in the West Midlands/England area studied with national trends.3.Assess ADR incidence locally versus nationally.4.Examine associations between semaglutide prescribing and socioeconomic deprivation.

## 2. Materials and Methods

### 2.1. Study Design and Ethical Approval

This was a retrospective, population-based, multicentre service evaluation. Data were collected from 42 general practices across the studied borough, West Midlands/England, for patients prescribed semaglutide between 2019 and 2024. Ethical approval was granted by the University of Birmingham Ethics Committee (Ref: MP2425 047, December 2024) and governance approval was obtained from the GP surgeries integrated healthcare ethics board (August 2024). Data were stored and analysed using Microsoft Excel; analysis included descriptive statistics and frequencies.

### 2.2. Real-World Patient Data

Anonymised patient-level data were extracted and compiled into a structured Excel database using a data share agreement across the respective practices. Variables included age, gender, ethnicity, semaglutide dose and prescription type, dates of first and last issue, HbA1c and weight measurements, GP practice, and reported adverse effects. Data cleaning and curation ensured completeness for inclusion in analyses.

For both HbA1c measurements and weight measurements, the starting point used for each patient was the latest measurement taken within the 6-month period before the first semaglutide issue. The end point used was the measurement taken within 6 months (before or after) of the most recent issue. Patient data were excluded if there was less than 1 month between their first and most recent issue; their HbA1c or weight were not measured within the 6-month period before their first issue; or their most recent measurement was more than 6 months before or after their most recent issue, as this was not deemed an up-to-date measurement. The mean and standard deviation were calculated for the start and end HbA1c and weight measurements for the whole patient population.

### 2.3. National Prescribing Data

National semaglutide prescribing data were collected from the OpenPrescribing [23] database from January 2020 to September 2024, with data from 2019 being omitted due to only being available from the month of September, therefore data for the year 2019 was incomplete. National prescribing trends were compared with the population in the area studied to identify similarities and deviations.

### 2.4. Adverse Drug Reaction Data

ADR data were sourced from the Medicines and Healthcare products Regulatory Agency (MHRA) Yellow Card Scheme’s interactive drug analysis profile (iDAP) [24] for the years 2020–2024. Data were analysed by system organ class. To calculate incidence, national prescription numbers (from OpenPrescribing) and local prescribing figures were used to derive ADR reporting rates.

### 2.5. Effectiveness Outcomes

HbA1c and weight were evaluated for treatment response. For each patient, the baseline was defined as the most recent value within six months prior to semaglutide initiation. The follow-up value was the closest measurement within six months of the final semaglutide prescription. Patients were excluded if data were missing, or if the interval between start and end measurements was <1 month or >6 months. The mean and standard deviation were calculated using Microsoft Excel.

### 2.6. Identification of Socioeconomic Impact

GP practices were assigned a deprivation ranking using the Socioeconomic Index for Small Areas (SEISA) decile from the Higher Education Statistics Agency [25]. Practice postcodes were mapped to SEISA deciles (1 = most deprived and 10 = least deprived). Prescription volumes were analysed across deciles to assess socioeconomic patterns.

## 3. Results

### 3.1. Prescribing Patterns

Between 2019 and 2024, semaglutide prescribing in the area studied (GP practices) increased initially, peaking in 2022 with 446 prescriptions, then declined sharply to 63 prescriptions by 2024. National prescribing followed a similar trend but showed a more gradual decline after March 2023 (Figure 1).

### 3.2. Ethnicity of Patients Prescribed Semaglutide

Among the 1403 patients prescribed semaglutide in the area studied, 82.5% identified as white, with White British patients accounting for 41%. The low number of White British patients prescribed semaglutide vs. 82% of the region population warrants further investigation. Other ethnicities, including South Asian patients, comprised 7.2% of those prescribed semaglutide and aligned with the local demographic (7.2%). Minoritised ethnic groups represented a small proportion of prescriptions, with some categories (e.g., South and Central American) accounting for a single patient. A total of 52% of patients were female and the age range was 20–91 years of age (Table 1).

The mean baseline HbA1c was 74.7 mmol/mol, falling to 63.9 mmol/mol, a reduction of 10.8 mmol/mol (~1.0% absolute HbA1c drop) (Figure 2).

### 3.3. Treatment Efficacy

Mean body weight decreased from 91.4 kg to 87.0 kg, representing a 4.8% reduction (Figure 3).

### 3.4. ADRs

From 2020 to 2024, 23 patients (1.85%) in the area studied reported semaglutide-related ADRs, compared to a national reporting rate of 0.21%. Nationally, gastrointestinal symptoms were the most common (54%), followed by general disorders (13.3%) and nervous system disorders (10.7%), (Table 2 and Table 3).

### 3.5. Prescription Type

Fifty-eight percent of patients transitioned from an acute to a repeat prescription, indicating ongoing use. However, 61% of patients had discontinued semaglutide by 2024. Among these, 24% received only acute prescriptions, with some receiving a single dose (Figure 4).

### 3.6. Socioeconomic Impact

Prescribing was highest in practices located in the most deprived areas (SEISA deciles 1–2), with the single highest-prescribing practice situated in a decile 1 area. Prescribing levels declined as deprivation decreased (Figure 5).

## 4. Discussion

This real-world service evaluation explored the prescribing patterns and adverse effect reporting of semaglutide in the region studied, a socioeconomically deprived region of England [22], and compared them with national data. The findings offer new insights into local variability in access, treatment continuation, and pharmacovigilance.

### 4.1. Prescribing Patterns and Supply Constraints

Semaglutide prescribing rose rapidly following its national approval, peaking in 2022, before declining in 2023–2024. This temporal pattern mirrors national trends but was more pronounced locally. The timing of this decline aligns with national GLP-1RA shortages and the 2023 NHS Patient Safety Alert [26]. This suggests that access to semaglutide is especially vulnerable to supply chain pressures in areas with high treatment demand and constrained healthcare resources. The findings also reflect broader challenges in balancing demand driven by clinical indications versus off-label and cosmetic use [9,11].

### 4.2. Ethnic Disparities in Prescribing

Data showed that 83% of semaglutide recipients in the area studied identified as white (incl. British and other) and 1.3% identified as mixed ethnicity. The data shows that both white British (82.4% of region but only 41.3% of Rx) and mixed ethnicity (2.8% of region but only 1.3% of Rx) experience lower prescribing compared to South Asian and Black patients. This may be due to the elevated prevalence of type 2 diabetes and cardiometabolic risk in Black and South Asian populations [27]. Other white and European patients received 11.8% of all prescriptions but represent only 2.4% of the ethnicity of the region. While this could reflect differences in prescribing or potential differences in tolerability or efficacy, it also reflects clinical decision making. Previous analyses have highlighted how trial underrepresentation and limited subgroup data contribute to uncertainty about treatment effectiveness in certain populations [27,28].

### 4.3. Clinical Effectiveness in Practice

The observed reduction in HbA1c (~11 mmol/mol) and mean weight loss (4.8%) are consistent with findings from the SUSTAIN and STEP trial programmes [7,8]. These results affirm the translational validity of semaglutide’s efficacy in routine UK primary care, even within a population facing elevated comorbidity and deprivation. However, the lack of dose-specific or duration-adjusted data in our cohort limits firm attribution. Moreover, no clear association was found between socioeconomic status and treatment response, suggesting that semaglutide is likely effective across deprivation gradients when access is ensured.

### 4.4. Underreporting of ADRs

The reported ADR incidence in the area studied (1.85%) was ninefold higher than the national rate (0.20%), yet both figures fall far below rates seen in clinical trials, where gastrointestinal symptoms affect up to 74% of patients [8]. This disparity highlights widespread underreporting, a known issue within passive pharmacovigilance systems [26]. Underreporting may result from low perceived severity, patient reluctance, lack of awareness, or limited guidance from healthcare professionals [26]. It is likely exacerbated in contexts where semaglutide is prescribed online or without adequate counselling, bypassing opportunities for ADR education [10,11,29]. Future pharmacovigilance efforts should explore the integration of automated prompts in electronic health records and proactive surveillance in community pharmacies and primary care.

### 4.5. Treatment Continuity and Medication Switching

Only 39% of the patients studied continued semaglutide treatment through the study period, with 61% discontinuing treatment. This may be partially explained by tolerability issues or clinical inertia following early discontinuation. However, this finding also coincides with the 2023 UK approval of tirzepatide, a dual GLP-1/GIP agonist, which demonstrated superior glycaemic and weight outcomes compared to semaglutide in the SURPASS programme [30]. The extent to which semaglutide discontinuation reflects planned treatment switching, treatment fatigue, or issues with access remains unclear. Future studies should investigate prescribing trajectories, switching behaviour, and patient-reported reasons for discontinuation.

### 4.6. Deprivation and Prescribing Volume

There was a clear association between deprivation and prescribing rates, with the highest levels of semaglutide use in practices located in SEISA deciles 1 and 2. This reflects known associations between socioeconomic deprivation, higher diabetes prevalence, and earlier onset of complications [31,32]. It also raises important questions about resource allocation, treatment prioritisation, and the capacity of deprived areas to maintain continuity of care during supply disruptions. Notably, our findings also challenge assumptions that newer therapies are disproportionately accessed by affluent groups, demonstrating that effective primary care networks in deprived areas can deliver equitable access when supported.

### 4.7. Recommendations to Practice

These findings have several implications. First, clinicians should be supported to routinely counsel patients on common and serious adverse effects, including how and when to report these using systems like the Yellow Card Scheme. Structured follow-up, particularly after initiating GLP-1RAs, could improve early identification of ADRs and reduce unnecessary discontinuation. Second, efforts to curb inappropriate prescribing—particularly the off-label weight-loss use of Ozempic—should be accompanied by public health messaging to reduce demand driven by non-clinical motivations and unregulated online supply. Third, prescribing should be monitored at the population level to identify disparities by ethnicity or deprivation, using routinely collected NHS data.

### 4.8. Strengths and Limitations

This multicentre study enlisted a large sample size of 1403 patients from 42 GP practices, conducted within an area of known deprivation in the UK where there are evident health inequalities [22] and where, to our knowledge, research of this nature has not previously been conducted. However, key limitations include patient data requiring exclusion due to incomplete data available, and no access to the nature of ADRs experienced by the studied patients, co-morbidities, or concomitant medicines prescribed. Therefore, it cannot be known with certainty if HbA1c and weight reductions were solely due to semaglutide treatment. Data extracted from the Yellow Card and OpenPrescribing databases were for semaglutide in general, including uses in other indications, e.g., weight management where typically higher doses are prescribed than for T2DM; therefore, the ADRs may not reflect the doses of semaglutide used. As with all pharmacovigilance studies interpreting data from national reporting systems, this study is affected by reporting bias, whereby data included from the MHRA Yellow Card system may have included self- reported false positive ADR reports and underreporting; therefore, data may not be reflective of real-life incidences of adverse reactions experienced with semaglutide.

### 4.9. Future Work

Future research should build on these findings by exploring ADR reporting behaviours among patients and clinicians, potentially through qualitative methods or surveys. Further analysis is needed to understand switching patterns to newer agents like tirzepatide and to examine long-term glycaemic durability. A broader study incorporating other localities in England would enable regional comparisons and generate more generalisable data. Expanding this analysis to include other GLP-1RAs would also help to identify class-wide or drug-specific trends in tolerability and adherence.

## 5. Conclusions

This research contributes novel insights into the patterns of semaglutide prescribing within the region studied in England and is the first study to compare these findings with national data to offer recommendations for current GLP-1RA prescribing practices in the UK. We have highlighted the prominent issue of underreporting of adverse drug reactions observed with semaglutide. We have illustrated the clear correlation between lower socioeconomic status and higher prescribing of semaglutide for type 2 diabetes in the area studied. In addition, considered the potential causes of the recent decrease in semaglutide prescriptions issued. The prescribing patterns in the area studied were in general agreement with national patterns, and the incidence of ADRs experienced was lower than expected both locally and nationally in comparison with the literature. However, further investigation into the nature of the ADRs experienced in this population could elicit additional conclusions such as differences between patient characteristics (e.g., ethnicity and age).

## Figures and Tables

**Figure 1 healthcare-14-00035-f001:**
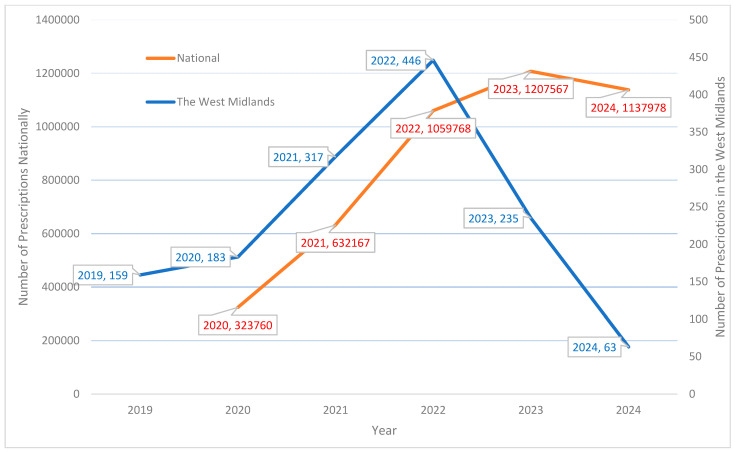
Annual semaglutide prescribing in the West Midlands/England area studied vs. nationally.

**Figure 2 healthcare-14-00035-f002:**
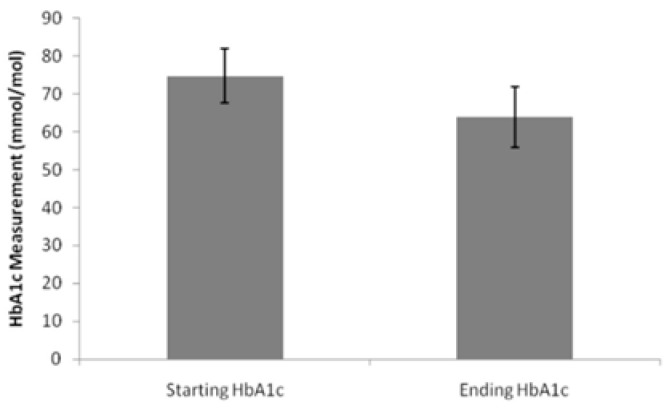
Mean changes in HbA1c.

**Figure 3 healthcare-14-00035-f003:**
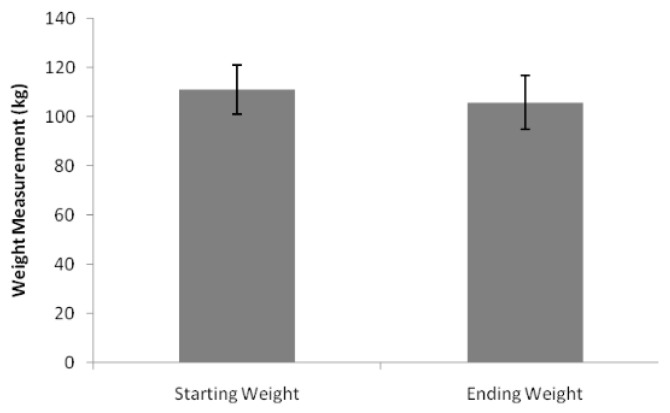
Body weight with standard deviation error bars.

**Figure 4 healthcare-14-00035-f004:**
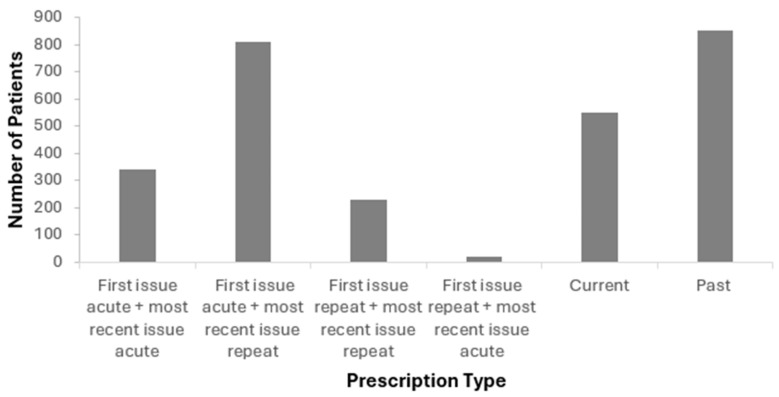
Initial and current prescription types and continuation rates for the area studied.

**Figure 5 healthcare-14-00035-f005:**
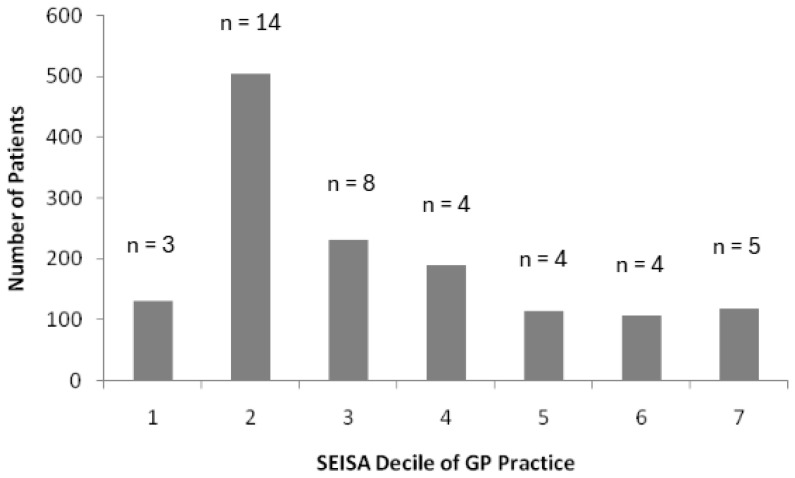
Number of patients prescribed semaglutide in the area studied based on SEISA decile of the 42 GP practices. n values represent the number of GP practices for that SEISA decile value. SEISA decile extracted for each GP practice from Higher Education Statistics Agency (HESA) [25].

**Table 1 healthcare-14-00035-t001:** Ethnic distribution of semaglutide recipients in the area studied. n/a = data not available.

Ethnic Category	Number of Semaglutide Prescriptions	% of Total Semaglutide Prescriptions	% Ethnicity Make-Up of West Midlands Region Studied (Census, 2021)
White (incl. British and Other)	1157	82.5	84.8
White British	580	41.3	82.4
British or Mixed British	412	29.4	n/a
Other White and European	165	11.8	2.4
South Asian	101	7.2	7.2
Black (African, Caribbean, or Other)	38	2.7	2.5
Mixed Ethnicity	18	1.3	2.8
Other (Asian, South/Central American)	32	2.3	2.5
Undisclosed	53	3.8	n/a
Total patients prescribed semaglutide	1403	n/a	n/a

**Table 2 healthcare-14-00035-t002:** Incidence of adverse drug reactions reported nationally versus in the West Midlands region studied between 2020 and 2024.

	Total Number of Adverse Drug Reactions Reported from 2020 to 2024	Total Prescription Number from 2020 to 2024	Incidence of Adverse Drug Reactions from 2020 to 2024 (%)
National ^(A)^	9429	4,500,312	0.21
The WM area studied	23	1244	1.85

^(A)^ National total prescription number is from OpenPrescribing database [23]. National adverse drug reaction value is from the MHRA Yellow Card site [26]. As data from OpenPrescribing for the year of 2019 was incomplete, the data from 2019 from OpenPrescribing, MHRA Yellow Card, and real-life patient data from the WM area studied was omitted to create a standardised comparison of incidence.

**Table 3 healthcare-14-00035-t003:** Summary of adverse drug reactions reported to MHRA Yellow Card system by organ class for semaglutide (from 2020 to 2024).

	Total Number of Reactions	% of All Reported Adverse Drug Reactions
Gastrointestinal disorders	5116	54.1%
General and administration site reactions	1259	13.3%
Nervous system disorders	1009	10.7%
Psychiatric, skin, and metabolism disorders	754	8.0%
Psychiatric	305	
Skin and subcutaneous tissue	216	
Metabolism and nutrition	233	
Injury, investigations, and procedural issues	540	5.7%
Injury/poisoning	275	
Investigations	250	
Surgical/medical procedures	10	
Other organ/system disorders	728	7.7%
Includes cardiac, hepatic, respiratory, infections, renal, endocrine, and vascular disorders.		
Fatalities	23	0.2%
Total number of ADRs	9429	

## Data Availability

The data presented in this study are available on request from the corresponding author due to ethical implications.

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
