# Peer review of "Prescribing Patterns and Adverse Effects of Semaglutide: A Real-World Comparative Evaluation"

_healthcare, 2025, doi:10.3390/healthcare14010035_

Round 1

Reviewer 1 Report

Comments and Suggestions for Authors

This research examined the patterns of semaglutide prescribing and adverse effects experienced by patients in the West Midlands area studied and correlated with those identified nationally. The study is well designed and well written in clear scientific manner.

I have only few minor suggestion.

The Results section should not state conclusions, but only the findings of the study. Therefore, the sentences or parts listed below should be deleted or moved to appropriate paragraphs in the Discussion.

135-136:  ‘’These changes support semaglutide’s real world effectiveness in improving glycaemic control and weight.’’

151-152: ‘’Despite higher local reporting, the overall rates remain well below those reported in clinical trials, highlighting underreporting.’’

159: ‘’Discontinuation may reflect side effects, efficacy concerns, or supply constraints.’’

168: ‘’’….suggesting a socioeconomic gradient in semaglutide use.’’

With those minor changes, the article may be accepted for publishing.

Reviewer 2 Report

Comments and Suggestions for Authors

I appreciate the authors for choosing the topic “Prescribing Patterns and Adverse Effects of Semaglutide: A Real-World Comparative Evaluation”. I believe this study addressed the Semaglutide prescribing inequalities in the UK population by comparing national data with regional data, supported by clinical evidence. I suggest numerous recommendations to the authors to improve the quality of the manuscript.

Abstract

Some abbreviations have no description on their first appearance, i.e., GLP-1, UK, HbA1c, and MHRA. So, I suggest that the authors add the description for the abbreviation on its first appearance.

Aims

“To examine real-world prescribing patterns and ADRs associated...”. The authors also report the treatment efficacy in their study. So, I suggest the authors rephrase the sentence as “To examine real-world prescribing patterns, treatment efficacy, and ADRs associated...”. 

Introduction

The abbreviation “GPhC” needs a description on its first appearance.

The introduction is brief and does not address the gap between the background and the research objectives with enough evidence. The authors need to conduct an extensive literature review and revise the introduction to provide more details about the current scenario.

“Yet pharmacovigilance data remain limited…” Please refer to the pharmacovigilance data from the VigiAccess and EudraVigilance databases of the WHO, which covers the global data. The authors should address the background by including more detailed real-world evidence related to off-label use and other outcomes in healthcare practices in the context of prescription practice. In this regard, the authors should address the need for this present research more explicitly.

https://doi.org/10.2174/0115748863367086250420011411

https://doi.org/10.3390/biomedicines12051124

“Furthermore, little is known about how prescribing varies by local demographics, ethnicity, and socioeconomic status.” The sentence is convincing. I recommend that the authors provide some evidence of what is already known, which helps readers understand the background well.

https://doi.org/10.1136/bmjmed-2025-001349

Lines 65, 67, 69, and 70: I suggest that the authors start the sentence with a capital letter.

Line 79: “(August(2024)”. Please change this to (August 2024)

Method

I checked the raw data by downloading the Excel file from https://openprescribing.net/chemical/0601023AW/ (Ref. 21) and found data up to July 2025. I think “..database from January 2020 until September 2024.” should be changed to “..database from January 2020 until July 2025..” Also, I did not find anything in the year 2019. Please clarify and rewrite the paragraph to be relevant to the data present in the reference. 21.

Line 93: The abbreviation ADR has already been described in Line 54, and please avoid duplication.

Line 93: MHRA needs its full form on its first appearance.

Lines 102-103: “Patients were excluded if data were missing, or if the interval between start and end measurements was <1 month or >6 months.”. Please provide more details about the exclusion and how many patients were excluded due to missing data, as well as those excluded for being <1 month or >6 months. Also, please address what type of data is missing in general?

Results

Figure 1: I believe the authors have collected data from the West Midlands; however, the Y axis in Figure 1 is labeled as “Dudley”. What is “Dudley”. Is it other than the West Midlands?. Please be consistent in the text and the Figure with similar text.

Line 115: “..then declined sharply to 60 “. I found in the Figure it’s 63. Please be consistent with the data in the figure and text.

Figure 1: On the Y-axis, left side. Please change “NAtionally” to Nationally.

Figure 1: 63 (2024). 63 prescriptions should be higher than the gridline 50. But the 63 came down to 50, and I guess it’s wrong. Please change the graph and ensure the graphical representations are consistent with the data.

Line 121: “Among 1403 patients, 82.5% identified as White”. This sentence is not clear and may confuse the reader. What do you mean by “1403”? I guess the total number of patients who are eligible for Semaglutide!

Table 1: There is a missing column of the total number of eligible patients for Semaglutide on the left side of “Number of semaglutide prescriptions”, which will help to interpret the results quite easily. Also, add another column to “% ethnicity make-up of West Midlands region studied (Census, 2021)” and provide the total number of eligible patients. So that readers can understand how many of them are eligible and how many of them received.

Table 1: Please add a row for the total number of prescriptions in Table 1.

Figures 2 & 3: The current results will not assist in making a decision. I suggest the authors compare the mean differences of HbA1C and body weight using a paired t-test for the solid finding.

Lines 143-144: The references 38 and 39 are not in chronological order. Please arrange them in chronological order.

Tables 2 and 3: The sum of the total number of reactions is 10718 in Table 2. However, the Total number of Adverse Drug Reactions Reported from 2020 to 2024 in Table 3, according to the authors, is 9185. Please repeat the addition and provide consistent data in both tables. Please add one row for the total number of ADRs in Table 2.

Tables 2 and 3: I suggest “Incidence of Adverse Drug Reactions reported nationally versus in The West Midlands region studied between 2020 and 2024.” should be Table 3. Summary of Adverse Drug Reactions reported to MHRA Yellow Card System by organ class for semaglutide (from 2020-2024) should be Table 4. The former offers an overview of ADRs, and the latter provides more insights.

Tables 1,2, and 3: Number of prescriptions 2556 (Sum of Semaglutide prescriptions; Table 1), which is less than the number of ADRs reported (9185; Table 3) during the study period (2020-24).

Figure 4 shows a duplication in prescription type as “First issue repeat + most recent issue acute”. Please correct it.

Figure 4: Please provide a clear description of each prescription type in the method section.

Figure 4 and the relevant text (lines 156-159) do not make any sense, as the text does not reveal the data of the figure. In this case, there is no need for Figure 4.

Reviewer 3 Report

Comments and Suggestions for Authors

The study describes an important and current topic in clinical practice, the use of semaglutide and other GLP-1 receptor agonists and the associated pharmacovigilance.

In general terms, the study is well designed and the article is well defined.

However, the results need improvement, both in terms of presentation/formatting and content, as well as methodology.

The statistical analysis should be described in the methodology.

Although sociodemographic data were collected, the study population is never characterised. Ethnic data are presented, but there is no characterisation of age, sex, and other variables mentioned in the methodology. 

For better understanding, the figures and tables should appear after the text that introduces them, and not the other way around as presented.

A summary of adverse reactions is presented at the global level, but the ADRs reported in the West Midlands area are not characterised. It would be important to check for differences in ADRs reported in a specific area and to understand whether there are differences compared to national figures.

Similarly, the discussion should be consistent with the results presented.

With these minor changes, the content of the work could be improved and elevated.

Round 2

Reviewer 2 Report

Comments and Suggestions for Authors

I appreciate the authors' efforts in revising the manuscript. However, the following comments have not yet been addressed in the method section. I found the following in their reply.

“Furthermore, little is known about how prescribing varies by local demographics, ethnicity, and socioeconomic status; however, existing studies have suggested there are potential inequities in prescribing that require further exploration. (ref) .” The sentence is convincing. I recommend that the authors provide evidence of what is already known to help readers understand the background.
https://doi.org/10.1136/bmjmed-2025-001349

Lines 102-103: “Patients were excluded if data were missing, or if the interval between start and end measurements was <1 month or >6 months.”. Please provide more details about the exclusion and how many patients were excluded due to missing data, as well as those excluded for being <1 month or >6 months. Also, please address what type of data is missing in general? ASK ALAN AND ZAHRA?

Author Response

Comment 1 and response- reference was added 

“Furthermore, little is known about how prescribing varies by local demographics, ethnicity, and socioeconomic status however, existing studies have suggested there are potential inequities in prescribing that require further exploration [22]

.” The sentence is convincing. I recommend that the authors provide some evidence of what is already known, which helps readers understand the background well.

https://doi.org/10.1136/bmjmed-2025-001349

comment 2- Lines 102-103: “Patients were excluded if data were missing, or if the interval between start and end measurements was <1 month or >6 months.”. Please provide more details about the exclusion and how many patients were excluded due to missing data, as well as those excluded for being <1 month or >6 months. Also, please address what type of data is missing in general?

response 2- For both HbA1c measurement and weight measurement, the starting point used for each patient was the latest measurement taken within the 6-month period before the first semaglutide issue. The end point used was the measurement taken within 6 months (before or after) of the most recent issue. Patient data was excluded if: there was less than 1 month between their first and most recent issue; theirHbA1c or weight were not measured within the 6 month period before their first issue; or their most recent measurement was more than 6 months before or after their most recent issue as this was not deemed an up-to-date measurement. The mean and standard deviation were calculated for the start and ending HbA1c and weight measurements for the whole patient population